# Experiences of the Telemedicine and eHealth Conferences in Poland—A Cross-National Overview of Progress in Telemedicine

Rafał J. Doniec [1,2,*], Natalia J. Piaseczna [1], Karen A. Szymczyk [1], Barbara Jacennik [2,3], Szymon Sieciński [1], Katarzyna Mocny-Pachońska [4], Konrad Duraj [1], Tomasz Cedro [2], Ewaryst J. Tkacz [1] and Wojciech M. Glinkowski [2,5,*]

1. Department of Biosensors and Processing of Biomedical Signals, Faculty of Biomedical Engineering, Silesian University of Technology, Roosevelta 40, 41-800 Zabrze, Poland
2. The Polish Telemedicine and eHealth Society, Targowa 39A/5, 03-728 Warsaw, Poland
3. Prince Mieszko I College of Applied Sciences in Poznań, Bułgarska 55, 60-320 Poznan, Poland
4. Department of Conservative Dentistry with Endodontics, Faculty of Medical Science, Medical University of Silesia, Pl. Akademicki 17, 41-902 Bytom, Poland
5. Center of Excellence "TeleOrto" for Telediagnostics and Treatment of Disorders and Injuries of the Locomotor System, Department of Medical Informatics and Telemedicine, Medical University of Warsaw, 00-581 Warsaw, Poland
* Correspondence: rafal.doniec@polsl.pl (R.J.D.); wojciech.glinkowski@wum.edu.pl (W.M.G.)

**Abstract:** The progress in telemedicine can be observed globally and locally. Technological changes in telecommunications systems are intertwined with developments in telemedicine. The recent COVID-19 pandemic has expanded the potential of teleconsultations and telediagnosis solutions in all areas of medicine. This article presents: (1) an overview of milestones in the development of telecommunications systems that allow progress in telemedicine and (2) an analysis of the experiences of the last seven conferences of telemedicine and eHealth in Poland. The telemedicine and eHealth conferences have grown steadily in Poland since their inception in the late 1990s. An exemplary conference program content was used to assess the scientific maturity of the conference, measured by the indices of research dissemination and the impact of publications. The overview presents progress in selected areas of telemedicine, looking at local developments and broader changes. The growing interest in telemedicine in the world's medical sciences is demonstrated by visibility metrics in Google Scholar, Pubmed, Scopus and Web of Science. National scientific events are assumed to raise interest in the population and influence the creation of general policies. As seen in the example of Poland, the activity of the scientific community gathered around the Polish Telemedicine Society led to novel legal acts that allowed the general practice of telemedicine during the SARS-CoV-2 pandemic. Local scientific conferences focusing on telemedicine research can be a catalyst for changes in attitudes and regulations and the preparation of recommendations for the practice of telemedicine and electronic health. On the basis of the results of this study, it can be concluded that the progress in telemedicine cannot be analyzed in isolation from the ubiquitous developments in technology and telecommunications. More research is needed to assess the cumulative impact of long-standing scientific conferences in telemedicine, as exemplified by the telemedicine and eHealth conferences in Poland.

**Keywords:** e-health; telemedicine; information systems; publication impact; COVID-19 pandemic; telehealth

## 1. Introduction

The COVID-19 pandemic period undoubtedly increased and accelerated interest in and use of telemedicine in healthcare [1,2] in various medical specialties as a safe and effective way to deliver healthcare [1–3]. Telemedicine ("tele" means at a distance, and

"medicine" refers to medicine, considered the art of healing) is a rapidly growing way of providing medical services remotely [4–8]. In its purest form, telemedicine refers to treatment by a physician who is professionally responsible for providing this service remotely to a patient without face-to-face contact [9–11]. Other medical services that do not necessarily include direct interaction with a physician are generally referred to as telehealth or eHealth [12–14]. Telehealth includes health services in the form of various applications [15]. An operational definition of telemedicine stimulates the discussion of its role in the healthcare system and the global economy [16]. Economic analysis of telemedicine, its basic approaches and requirements for evaluating and identifying core issues require further research around the world as healthcare delivery environments evolve due to rapid technological advances [16,17]. For this paper, we adopted a research hypothesis that conference speeches were related to subsequent publication of scientific papers. This work aimed to look at telemedicine in Poland from two perspectives. On the one hand, it reviewed the conditions and progress in telemedicine, i.e., the development of the telecommunications system. On the other hand, the corpus of 45 presentations from telemedicine and eHealth conferences in Poland was analyzed to assess the dissemination of research and the impact of publications facilitated by the Polish Telemedicine Society.

## 2. Materials and Methods

The methodology of this study focused on the paths that supported the development of telemedicine in the Web 2.0 period (that is, the digital transformation of innovation after the dotcom crash of 2001) [18–20]. Furthermore, the methodology considers the role that Polish telemedicine and eHealth conferences and changes in global telemedicine play in the prospects of developing telemedicine in Poland [4]. Speeches on telemedicine at the Telemedicine and e-Health Conference 2014 were assessed regarding their impact on subsequent publications. The presentations at this conference (titles and authors) served as input for the advanced Google Scholar search for the post-conference period until June 2021. The study was carried out based on the Google Alerts search and the list of e-sources available in the resources of the Digital Publishing House of the Silesian University of Technology. All titles that returned specific citations were considered positive and listed in the references. The number of positive responses was added as a related finding. The SQL JOIN function was used to select and compare the names of the first two authors and more than 50% of the characters.

Various areas of telemedicine were investigated, and scientific publications based on Google Scholar were searched. The impact of the presentations was expressed in subsequent follow-up published articles. This assessment of the impact of presentations and papers delivered at the Polish Society for Telemedicine and eHealth Conference, 2014, on the general field of telemedicine can therefore be treated as a novelty of our approach [5].

The overview approach considers phenomena that affect the development of telecommunication networks and technologies. One of the measures is the number of phone users per 100 inhabitants in Poland. Statistics published by the Office of Electronic Communications (UKE) and the Google browser were used.

It was necessary to create a standard query language (SQL) that compares the agreement of the first two authors of the conference presentation and, as a text variable, to compare conference papers and publications in scientific text databases. Similarly, elements and variations of titles being searched were queried in scholarly databases of scientific texts and compared, excluding conjunctions and punctuation marks. The percentage of publications related to presentations can reliably express the scientific maturity of presentations given at a scientific conference. Scientific search engines such as Google Scholar, Pubmed, Scopus and Web of Science were used to search and confirm publications related to the presentations.

Furthermore, the statistics of the World Bank of the International Telecommunications Union (ITU) for 1989–2020 [21,22] were compared and analyzed to understand the technological environment necessary for the growth of modern medical research based on new telecommunication technologies (see Figure 1).

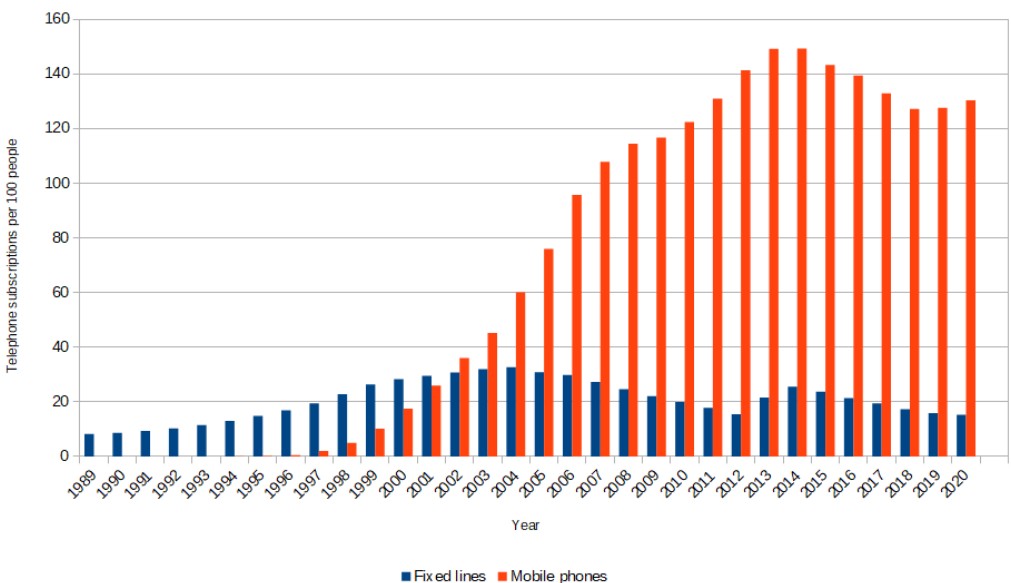

**Figure 1.** Number of telephone users per 100 inhabitants in Poland between 1989 and 2020.

## 3. Results

Poland broke free from the Eastern Bloc in 1989. At the same time, the era of isolation from Western culture and technology ended. Communist ideology was replaced by a free-market approach. It has finally become possible and necessary to rebuild the intellectual capital of the reborn sovereign state. At that time, however, there was only one national telecommunications operator in Poland, Telekomunikacja Polska SA (TPSA). In 1991, the development of mobile telephone systems began [23] with two analog mobile systems. In June of that year, Ameritech and France Telecom each received a 24.5% stake in a venture called "Polska Telefonia Komórkowa" (PTK). At that time (in 1991), Western bans on importing advanced technology to Poland were still in force. However, Poland's accession to the European Union removed these restrictions. In June 1992, PTK launched a brand assigned to the Centertel mobile network in Warsaw. Telekomunikacja Polska was also privatized and split into TPSA to handle telecommunications and Poczta Polska to provide postal services while retaining 100% of the government's shares.

As a result of privatization, in 2000, TPSA lost its monopoly on long-distance services, and at the end of 2002, it lost its monopoly on international calls [24]. The crucial considerations were reported in the "Report on the state of the telecommunications market in Poland in 2021", published by the Office of Electronic Communications (UKE). According to this report, Poland ranked last in the European Union for fixed-line Internet penetration (22.4%) [25]. In terms of the range of Internet access, assuming that the number indicated by the Google search engine has no territorial boundaries, it should be assumed that we have the same range as the world. Despite the tremendous technological leap that has occurred in Poland, the presented report still points to technological shortages in Poland in the form of state-of-the-art telecommunications and satellite links.

Research on the development of mobile telephone networks, the main carrier for telemedicine services, show the changes in the environment for telemedicine.. However, the concept of telemedicine was already developing in the medical community well before the breakup of TPSA. A family physician was published a paper on the prospects of telemedicine in Poland in 1996 [26]. In 1997, the Polish Society of Telemedicine was founded.

In 2004, PTS coorganized the first conference with the Center of Excellence, "TeleOrto" for Telediagnostics and Treatment of Disorders and Injuries of the Locomotor System, Medical University of Warsaw. The TeleOrto Center of Excellence was established on the basis of the results of a competition conducted by the Ministry of Science and Information Technology in 2004. The application to establish the TeleOrto Center was among 100 similar applications (of 400 submitted) that received a positive evaluation from the Ministry. In 2005–2007, the Center for Information Systems in Healthcare co-organized the telemedicine conferences with PTS. In 2014, a joint conference was held with the European Society for Medical Imaging Informatics (EUSOMII). Since 2014, annual conferences have adopted the name and formula of the Telemedicine and e-Health Conference, and the conferences have become international. The PTS website was launched [27]. Since then, many healthcare systems have joined telehealth and healthcare information exchange networks to improve connectivity and broaden their horizons to provide adequate care to all. Table 1 presents the program of the National Conference on Telemedicine and eHealth (2014) as an example of a more detailed analysis of the interest in telemedicine.

**Table 1.** Telemedicine and e-Health conference program—edition 2014.

| Session | Name of the Presenter and Presentation Title |
| --- | --- |
| Plenary session | M. Kędzierski—Health Care Information in Poland<br>F. Sicurello—Telemedicine and Telecare for wellbeing and personalized cure [28]<br>J. Niedziałek—Research Funding in the famework program Horizon 2020 |
| Systems and applications | R. Mężyk—e-Health in Świętokrzyskie Voivodeship<br>K. Wołk—Telemedicine as a special case of the Machine Translation [29]<br>P. Masiarz—Regional Medical Information Systems—an example of the Świętokrzyskie Voivodeship<br>Ł. G. Szepioła—Measurement and Control System in Telemedicine applications<br>E. Brzozowska—Telemetric recording and analysis of uterine contractions and fallopian tubes [30]<br>R. J. Doniec—Online daily record of changes DiabLab and other methods of monitoring of diabetic blood glucose concentration<br>J. Sierdziński—Use of GIS tools in systems of telemedicine and e-Health<br>M. Plechawska-Wójcik—Evaluation and comparison of medical applications functionalities using the experts' method [31] |
| In service for elderly and disabled subjects | M. Bujnowska-Fedak—The use of telemedicine services in a care of older people with particular regard to their attitudes, needs and expectations in this regard [32]<br>K. Popławska—Mobile application for detection of falls on devices based on Android<br>K. Walesiak—Telerehabilitation in orthopedics [33]<br>A. Żukowska—Telepulseoksymetry during exercise in elderly patients<br>A. Czyżewska—Remotely supervised rehabilitation of patients with osteoarthritis of the hip<br>K. Krawczak—Satisfaction level of patients' who used the telerehabilitation platform [33]<br>Ł. Markiewicz—Telediagnostic software for measurement and classification of spinal curvatures and vertebral fractures |

**Table 1.** *Cont.*

| Session | Name of the Presenter and Presentation Title |
|---|---|
| Electronic documentation and economics | M. Karlińska—Measurement and evaluation of stages of electronisation of medical records<br>B. Pędziński—The status of implementation of electronic medical records in primary healthcare institutions in Poland [34]<br>J. Sierdziński—Costs and benefits of telemedicine solutions—methodology and results [35] |
| Sponsored | Asseco—Telemedicine overcomes geographical barriers due to new technology<br>CGM—Telemedicine—conditions of development in Poland |
| Commission of Informatics and Telemedicine of Polish Cardiac Society | M. Grabowski—Decision support systems and automatic diagnostics and algorithms in Cardiology<br>P. Balsam—Cost effectiveness of telemedicine solutions—scientific review of the literature [36]<br>Ł. Kołtowski—"Ideal" mobile medical device—from the standpoint of patient and physician [37] |
| International cooperation | E. Szkiłądź—Medicine and eHealth in the Program Horizon 2020<br>TBD—Momentum-European Project<br>E. Krupinski—Telepathology—the impact of color display calibration on interpretation accuracy and efficiency (videoconference with USA) [38–41] |
| EuSoMII & ISfTeH Meeting | F. Lievens, M.Jordanova—The Role of ISfTeH in the World of Telemedicine/eHealth [42]<br>E. Neri—EuSoMII—perspectives of international cooperation [43] |

Selected and marked with assigned citations, presentations in the following years gave rise to print publications within a few years after the conference. It concerned those works that could be related thematically and through characters' strings, which appeared after the conference. Based on these works, a particular scientific maturity of the conference presentations was indicated. Some of the papers presented were derived from already published papers and were not included in this list [44–47]. Eventually, sixteen presentations delivered at Telemedicine and eHealth 2014 found related equivalents in publications. Therefore, the calculated maturity factor for scientific concepts amounted to sixteen publications of the total 31 presentations (51.61%), which can be considered significant and very satisfactory to the organizers.

Furthermore, as an indicator of interest in telemedicine, we conducted various searches on the Internet with relevant terms "e-health" and "telemedicine"—comparing the number of selections in 2016 and currently (see Table 2).

On 2 December 2014, Web 2.0 in Europe was officially launched in Brussels. The European Telemedicine Deployment Plan was created to help organizations implement remote healthcare services using information technology. According to this roadmap, eighteen critical success factors for the implementation of telemedicine were provided with details, context, indicators, descriptions and case studies. At this time, it should be emphasized that the number of so-called calls "to the subscriber" increased in Poland five times. The number of publications on Internet portals increased in 2016–2022 by 28 times. Interest in eHealth issues expressed by the presence of websites searched by Google browsers in 2016–2022 increased from 1,220,000 to 34,300,000. The result represents a 28.11-fold increase in interest in eHealth issues over the past 6 years.

The term "telemedicine" was searched 152,270,000 times on Google from 2016 to 2022, while in 2016, there were 1220000 searches. The 6-year period covering the period of the COVID-19 pandemic resulted in a 209.59-fold increase in interest. Furthermore, the number of scientific publications available through the Pubmed search engine has increased [48–50].

**Table 2.** The results of search engine selections on the terms "eHealth" and "telemedicine" in 2016 vs. 2022.

| Term | Source | Year | Results |
|---|---|---|---|
| eHealth | Google | 2016 | 1,220,000 |
| eHealth | Google | 2022 | 34,300,000 |
| Telemedicine | Google | 2016 | 730,000 |
| Telemedicine | Google | 2022 | 153,000,000 |
| eHealth | Pubmed | 2016 | 26,744 |
| eHealth | Pubmed | 2022 | 60,123 |
| Telemedicine | Pubmed | 2016 | 24,428 |
| Telemedicine | Pubmed | 2022 | 53,548 |
| eHealth | Scopus | 2016 | 4487 |
| eHealth | Scopus | 2022 | 13,551 |
| Telemedicine | Scopus | 2016 | 28,118 |
| Telemedicine | Scopus | 2022 | 63,121 |
| eHealth | Web of Science | 2016 | 4785 |
| eHealth | Web of Science | 2022 | 11,788 |
| Telemedicine | Web of Science | 2016 | 14,211 |
| Telemedicine | Web of Science | 2022 | 39,292 |

The increase in the number of published articles with the keyword eHealth based on Pubmed was 2.25 times, 3.02 on Scopus and 2.46 on Web of Science. For the keyword telemedicine, the numbers were 2.19, 2.24 and 2.76, respectively. At subsequent conferences, the number of speeches at the Polish Telemedicine and eHealth conference (in 2014–2020) remained at a similar level, approximately 30.

## 4. Discussion

Some authors date the beginnings of telemedicine to the 19th century [51]. A milestone is considered to be the establishment of the microwave video network that linked Massachusetts General Hospital and Boston's Logan Airport by Kenneth Byrd and several other physicians in 1968. Terrestrial telephony in Poland (Warsaw) began operating on 13 July 1882 and was launched by the Bell International Telephone Company. A milestone in the development of telemedicine in Poland was the remote transmission of an electrocardiogram. It was carried out by Professors Marian Franke and Witold Lipiński over a distance of about 500 m in Lviv before World War II in 1935 [52]. A more comprehensive introduction of telemedicine solutions in Poland had to wait until the early 1990s [4]. In the 1990s, the first frequencies for mobile networks were launched in Poland [23]. Today, the number of telephones per capita has increased to approximately 1.36 per person.

In addition to the telecommunications infrastructure, an important background factor for the development of telemedicine is the emergence of very influential organizations and initiatives focused on telemedicine, often with a global dimension, including the National Telemedicine Initiatives initiated by the National Medical Library (NLM) [53], the Telemedicine Information Exchange (TIE) [54], ATSP—Association of Telehealth Service Providers [55] and ATA—American Telemedicine Association [14,56,57].

The mechanism for financing projects and initiatives in telemedicine that was launched influenced the development of telemedicine in its current incarnation. In October 1996, NLM announced funding for 19 multi-year telemedicine projects to serve as models for assessing the impact of telemedicine on cost, quality and access to healthcare, evaluating dif-

ferent approaches to ensuring the confidentiality of health data transmitted over electronic networks and testing emerging health data standards. The telemedicine projects awarded prioritized healthcare for the elderly, supported rural primary care physicians, supported newborn care facilities and provided health information to healthcare professionals in rural and urban areas.

TIE is located in all countries that have adopted the principles of telemedicine and is a comprehensive, international, quality-filtered resource center for information on telemedicine, telehealth and related activities. It is supported and maintained by ATSP and provides a global e-health information exchange.

ATSP is an international non-profit trade organization whose membership aims to improve healthcare through the development of the telehealth industry [56]. It was founded in 1996 by Douglas Perednia, MD, in Portland, Oregon. The association strongly believes that telemedicine is a practical tool that can improve the distribution of healthcare services for the benefit of patients and service providers.

ATA is an international leader in the promotion and advocacy of advanced remote medical technologies. ATA is working to integrate telemedicine into transformed healthcare systems to improve the quality, equity and affordability of healthcare worldwide. It was founded in 1993 as a mission-oriented non-profit organization headquartered in Washington, DC. The ATA's mission is to promote professional, ethical and equitable improvements in healthcare delivery through telecommunications and information technology [57,58].

The Polish Telemedicine Society (PTS) is a non-governmental association aiming to promote and develop multifaceted telemedicine and eHealth [59,60]. The first meeting of the founding members of the association under the name Polish Telemedical Society was organized on 22 May 1997 in Warsaw. The association has undertaken many initiatives, including the organization of annual scientific conferences and the co-organization of several international conferences. The association's achievements have been repeatedly presented at international and national conventions and conferences. Books of abstracts and full papers from many conferences have been published. Along with the design and implementation of the Association's decorations, such as the silicon badge for members of the association and the bronze, silver and gold "Bene Meritus" badges, the PTS Awards Committee was established to express appreciation for the merits of the development of telemedicine and eHealth in Poland and the world [61].

Many newly created telemedicine solutions need to be noted. The development of telemedicine, using various telecommunication methods was developed and implemented with remarkable speed due to the outbreak of the SARS-COV-2 pandemic [62–66].

The growing demands of users with powerful mobile devices and the growing need for remote medical services are giving developers new motivations to create applications that take advantage of an increasingly connected environment. It is vital to encourage software developers to create new ways to provide people with easy access to high-quality medical information, remote diagnostics and treatment and remote healthcare [67].

The interdisciplinary communication platform for programmers, software developers, telecommunication specialists, healthcare professionals and others was created at PTS's telemedicine and eHealth conferences. The impact of organized conferences may last longer in the form of the following publications that influence the further development of telemedicine. The related published papers may also prove the scientific maturity of the presented conference.

The comparison of similarities and shared spaces for issues raised during the Polish editions of the PTS-organized telemedicine and eHealth conferences was carried out due to the extensive introduction and explanation of the environment for the development of telemedicine in Poland.

The overview of telemedical topics addresses some similarities with the educational process. One concept that seems helpful in explaining the evolution of remote medical communication is the Coldeway quadrant [68] (see Figure 2). Remote visits can be compared to remote learning concepts called Coldeway quadrants (see Figure 3). According to them, there are four ways in which distance education can be practiced. The determinants are time and place, which may be the same or different. Education at the same time and place (ST-SP) is traditional classroom education. Compared to medical practice, it is a face-to-face visit between the patient and the physician. The second variant is different times, but the same place (DT-SP). It means that education takes place in an educational center, or students can attend classes in the same place but at a time of their choosing. The third variant takes place simultaneously, but at a different location (ST-DP), which means that telecommunication systems are used. Teleconferences or chats are used to connect students in different places simultaneously. This type of education is called synchronous distance education, which allows students to communicate in real time. Medical services are teleconsultations at a distance. Variant four means different times and places (DT-DP), the purest form of distance learning. Teachers and students can communicate asynchronously, at different times. However, in the case of telemedicine, despite the economic advantages, the asynchronous form is one of the least satisfactory for patients and physicians.

| Traditional education | Learning center on remote<br>Synchronous Distance Education (tele-education) |
|---|---|
| **Same time - Same place** | **Same time - Different Place** |
| **Different Time-Same Place** | **Different Time - Different Place** |
| Asynchronous Learning Center Education | Asynchronous Distance Education (tele-education) |

**Figure 2.** Coldeway's quadrants in distance education. Adapted from Dan Coldeway, while at Canada's Athabasca University [69].

| Traditional visit or short distance (in house) telemedicine based 24 hours in house patient monitoring | Telemedicine center/Synchronous Televisit or Teleconsultation<br>(phone or videoconference with or without sensors or other health status remote-operated detectors) |
|---|---|
| **Same time - Same place** | **Same time - Different Place** |
| **Different Time-Same Place** | **Different Time - Different Place** |
| Asynchronous medical observation or teleconsultation of the patient's data, including medical images, laboratory findings, and recorded interviews in the same medical facility. | Asynchronous medical observation or teleconsultation of the patient's data, including medical images, laboratory findings, and recorded interviews on the remote. Asynchronous procedures during the treatment, including self-assessment of health status with remote automatic anamnesis (medical history) through an application supporting the final diagnosis of the physician |

**Figure 3.** Modified version of Coldeway's quadrants for telemedicine.

The relationship between education and distance learning has recently become close in medicine. Tele-education has recently becomes a part of all medical specialties [19,70–73]. Continuing medical education using information and communication technologies (ICT) is flourishing among health professionals who need to attend to document their participation in training [71–74]. Depending on the needs, training of all medical and administrative personnel is necessary.

## 4.1. Telemedicine Training and Education

Polish laws and regulations have improved the physician's ability to communicate virtually with the patient to practice, make joint decisions, report progress safely and effectively and receive appropriate compensation for the services rendered [75]. Despite

this, teaching telemedicine and e-Health in pre- and postgraduate medical programs in universities remains marginal [76–82]. Medical students themselves demand that teaching of practical telemedicine be included in the curriculum. According to the law, medical studies still offer minimal knowledge and skills in telemedicine [79,83–85]. Although telemedicine services have ceased to be a theory, and with the pandemic, have become necessary, they have often become a practical lifeline for patients isolated during quarantine. The lack of training time adequacy results from teaching a subject containing telemedicine elements in the early years of studies. The training covers the youngest medical students who have not yet learned the methods of essential medical examination, let alone the examination modified for the needs of telemedicine [80,86]. In the curricula of medical universities, only the scientific foundations of medicine contain a provision that, in terms of knowledge, the graduate knows and understands the possibilities of modern telemedicine as a tool supporting the work of a physician [77]. However, the curriculum does not mention operative and non-surgical clinical science skills. Adopting the Act on the Medical Profession explicitly assumes the equivalence to the law and professional liability of providing medical advice in person and through ICT systems. Such provisions result in a gap in the teaching of the skills of performing medical consultations, conducting an adequate medical examination, collecting anamnesis and evaluating the results of additional examinations and medical images at a distance [87,88].

### 4.2. Telemedicine Technologies

Various technologies are involved in the provision of telemedicine services [63,89]. There is a clinical feedback loop between the physician and the patient, which determines the fulfillment of the actual condition of clinical telemedicine.

Telemedicine programs are inherently complex compared to their traditional on-site healthcare counterparts. Despite the necessity of the pandemic, developing sustainable multispecialty telemedicine programs is difficult. Single-service programs, such as teleradiology programs, are standard. Providing integrated telemedicine systems requires interrelated interoperability of telemedicine solutions with adequately efficient infrastructure. Several factors are barriers to developing integrated telemedicine programs [89].

Technological aspects that should be considered are tools, web services and ongoing service support. Currently, telehealth services enable 24/7 access to healthcare professionals through teleconsultations and elements of telecare and telemonitoring. Information and Communication Technologies (ICT) have great potential to support cost-effective and high-quality healthcare services. TM uses ICT to overcome geographical barriers, increase access to health services and increase the safety of patients and medical personnel in the face of the risk of microbial transmission and exceptional virulence, as in the case of the SARS-CoV-2 virus. Organizational issues of telemedicine also help with other tasks in the infrastructure of the healthcare provider, enabling: electronic appointments, virtual visits with a healthcare professional, access to health references via the Internet and using smartphone apps to track and monitor health statistics.

Telemedicine solutions also depend on whether a sufficiently efficient infrastructure is available for its needs. Telemedicine is becoming one of the most important ways to provide care today and in the future. Adapting to telemedicine solutions and constantly training physicians to use new telemedicine technologies for further development is necessary. Telemedicine networks [6,14,90,91] can provide services in the one-to-one (point-to-point), one-to-many (point-to-multipoint) and many-to-many (multipoint-to-multipoint) models. Initially, the telemedicine system was adopted to allow the patient to interact with a single physician, and the benefits cover many fields of teleradiology, teleneurology and others. Telemedicine communication models are depicted in Figure 4.

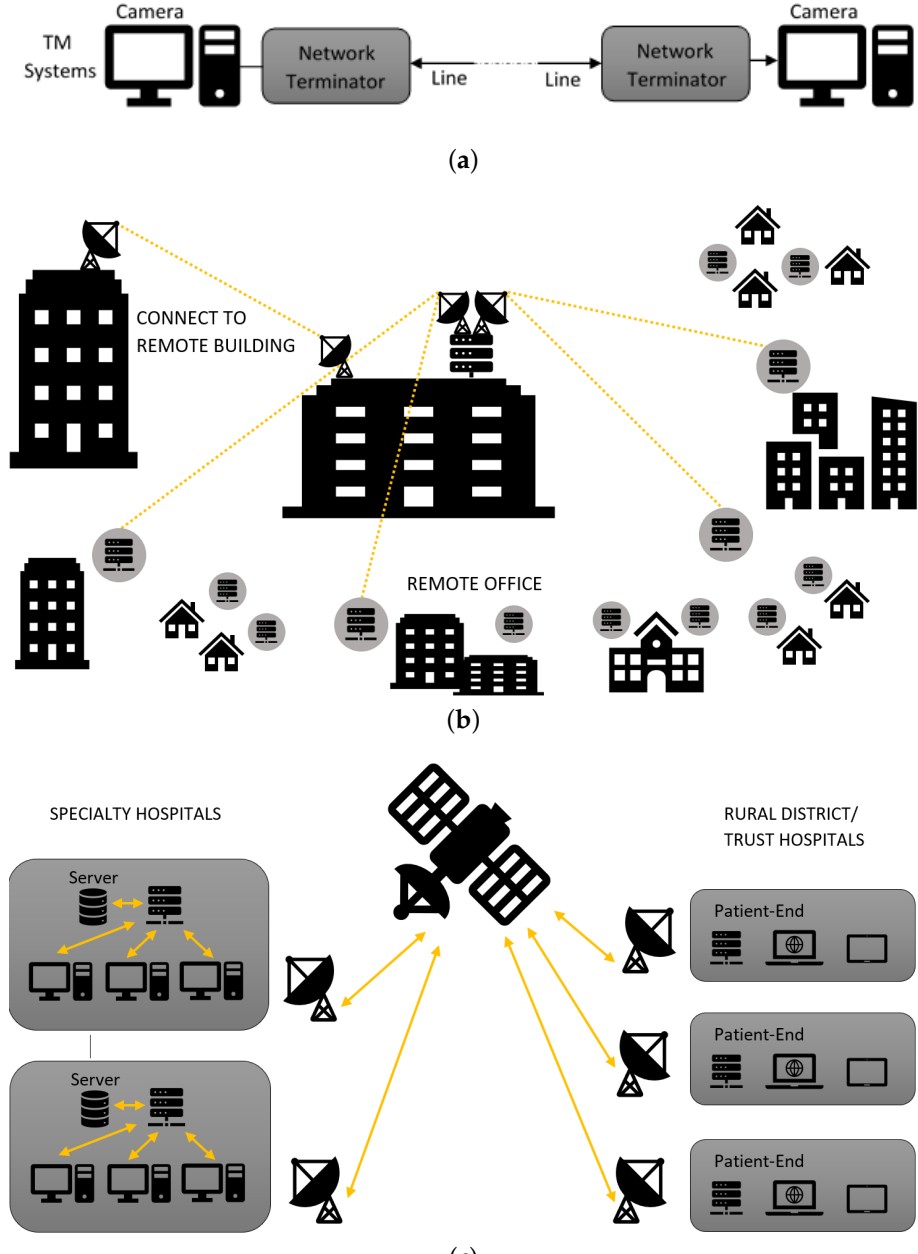

**Figure 4.** Schemes of the telemedicine systems. (**a**) Point-to-Point. (**b**) Point-to-Multipoint. (**c**) Multipoint-to-Multipoint.

### 4.3. Patient–Physician Communication in Telemedicine

Store-and-forward technologies are the electronic transmission of pre-recorded medical data (x-rays, video clips, scans, photos) between primary care providers and medical staff [92–96]. Store-and-forward methods reduce costs and improve access to healthcare services. However, the lack of real-time contact does not positively affect the quality of telemedicine services. The telephone retains its value in providing teleconsultations [97–102].

Videoconferencing allows the patient to consult with a physician in the comfort of their home with the possibility of seeing the physician face-to-face [103–105]. Almost all fields of medicine benefit from this type of real-time interaction. The only inconvenience is the lack of personal contact with the physician. However, it is possible to enrich the video transmission with reading from many diagnostic tools. Devices such as otoscopes, stethoscopes and other diagnostic devices can be connected to a computer, thus helping interactive examina-

tions [106]. However, real-time audio–video streaming requires a modern computer and network equipment with fast and low-latency Internet access, both in healthcare facilities and at the patient's place. The essential condition of medical devices used on the patient's site is the knowledge and skills necessary to use the devices. Inserting or applying a special probe dedicated to remote use often requires skills and training for safe use.

*4.4. Technologies in Telemedicine*

The constant development of modern technologies is quickly reflected in telemedicine in the form of many new tools and solutions such as artificial intelligence [107–110], wearable medical devices and sensors [111–116], health-related Internet of Things (IoT) [117–119], implantable medical devices operated by telemedicine, augmented reality [120–125], virtual reality [124,126–129] and support medical decision making [108,130–132]. Changes in physicians' attitudes toward the use of telemedicine solutions devices lead to changes in medical practice [133,134]. Altered medical practice leads to medical economy savings by delegating specific medical procedures to the point of care instead of the hospital.

The significant influx of data from medical devices related to telemedicine generates a considerable amount of medical data. Vast medical data sets require further research on medical pattern extraction to classify standardized features organized in data storage and related to the separation of ethical and data protection aspects. The development of mobile applications in recent years has increased the health awareness of users and enabled the collection of measurements from surrounding devices; they also act as proxies for powerful online "cloud" platforms that support artificial intelligence (AI) and "big-data" algorithms. Wearable and personalized solutions are created to support remote reading from miniature, battery-powered devices with built-in intelligent algorithms, sensors and wireless connectivity to provide measurements and results that are quickly available for monitoring, review and analysis. In Poland and worldwide, constant growth trends are observed for telemedicine (hospital and outpatient use) and the eHealth and mHealth market (consumer products) [135–139]. 'Start-ups' implemented by specialist teams create and deliver global solutions to specific problems in various medical specialties [30,31,45,140–142].

Telemedicine offers many benefits to patients, physicians and healthcare facilities. During the telemedicine and eHealth conferences, several trends were presented.

Thanks to the global reach of telemedicine and its network technologies, telemedicine gives patients access to physicians and allows healthcare facilities to extend services beyond their offices. With a global shortage of service providers, telemedicine can offset service shortages, increase patient throughput and positively impact the lives of millions of new patients in primary and secondary care. Implementing telemedicine can reduce costs, especially as healthcare systems have undergone point and monetary reform. Many countries have developed and implemented policies to reimburse costs and capital expenditures for services. The financial compensation model varies by country, region, state and telemedicine model [47].

Telemedicine currently does not differ from face-to-face medicine in terms of patient satisfaction. Over the past few years, numerous studies have documented a high level of patient satisfaction [137,143–145]. The easiest way to obtain the legality of telemedicine services is when there is equal responsibility for using stationary and remote services. In Poland, all telemedicine services are based on legal regulations [75,89]. However, they remain for cross-border solutions in telemedicine which require further studies.

Similarly, the topic of cybersecurity in telemedicine should be discussed separately. Modern cyberattacks exploit applications, networks, operating systems and hardware vulnerabilities. The most common purpose is to access, steal, or change sensitive information. Leakage or tampering with any patient's health can have dramatic consequences (e.g., drug or dose change, life-threatening surgery). In recent years, cybercriminals have shown no mercy even to hospitals, carrying out ransomware attacks that have stolen medical data and dysfunctions of medical systems and equipment, leaving medical staff and patients blackmailed to extort money to restore normal functioning.

## 5. Conclusions

The research allowed us to trace and demonstrate the effect of conference speeches in their scientific maturity expressed in publications directly related to them. The narrative of the review referred to the selected achievements of telemedicine and eHealth in Poland and showed the whole phenomenon in the environment of the development of telecommunications. Attention was also paid to the technological and historical environmental aspects of the challenge of telemedicine. This overview also referred to current challenges, policies and solutions in countries where innovations in the development of telemedicine are widely introduced. The pandemic period undoubtedly increased and accelerated the interest and use of telemedicine as a safe and effective path to healthcare delivery.

Telemedicine and information technologies are vehicles, methods and tools that enable the provision of health services but are also part of the process. However, broadband telecommunications and videoconferencing are available at the service of a qualified and experienced healthcare professional. It can be assumed that the assessment of the scientific effects of the impact on further publication achievements, using a global search engine, showed the importance of local activities on the example of the annual scientific conference, telemedicine and e-Health in Poland. The systematic and constant progress of of telemedicine has been presented against technological changes with telecommunications in the background.

The maturity of the publication of the presented papers turned out to be high in the case of the selected conference. The description of the environment of technical progress in telemedicine in Poland indicated important areas of telemedicine development, such as the activities of international and national organizations, education, technology, cybersecurity and the impact of local development on global impact in terms of the importance of telemedicine. The incredible growth of interest in telemedicine in the world's medical sciences is visible in Google Scholar, Pubmed, Scopus and Web of Science. Local initiatives and seemingly insignificant events can become milestones in promoting awareness. The need to use telemedicine becomes the proverbial "drop that trembles on the rock" and influences attitudes, regulations, conclusions, recommendations and practical solutions in telemedicine and ehealth. The phenomena of telemedicine development cannot be analyzed in isolation from the ubiquitous development of technology and telecommunications. More research needs to be conducted to assess the cumulative impact of multiple organized science events.

**Author Contributions:** Conceptualization, R.J.D., K.A.S., W.M.G.; methodology, R.J.D., S.S., W.M.G.; software, R.J.D.; validation, R.J.D., N.J.P., T.C., B.J. and W.M.G.; formal analysis, R.J.D., N.J.P., B.J., K.A.S., S.S., K.M.-P. and W.M.G.; investigation, R.J.D. and S.S.; resources, R.J.D., S.S.; data curation, R.J.D., T.C.; writing—original draft preparation, R.J.D., K.A.S.; writing—review and editing, R.J.D., N.J.P., B.J., T.C., K.M.-P., S.S., K.D. and W.M.G.; visualization, R.J.D., N.J.P.; supervision, R.J.D., N.J.P., E.J.T.; project administration, R.J.D.; N.J.P., E.J.T.; funding acquisition, R.J.D., E.J.T. All authors have read and agreed to the published version of the manuscript.

**Funding:** This research received no external funding.

**Institutional Review Board Statement:** Not applicable.

**Informed Consent Statement:** Not applicable.

**Conflicts of Interest:** The authors declare no conflict of interest.

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
