# Peer review of "Experiences of the Telemedicine and eHealth Conferences in Poland—A Cross-National Overview of Progress in Telemedicine"

_applsci, doi:10.3390/app13010587_

Round 1

Reviewer 1 Report

Authors give as a well-done description of telemedicine evolution and sincerely I think that the paper not need several changes to be able for publication.

Anyway, I suggest some changes that I think could improve the manuscript.

1)I fatigue to find a clear description of the scope of the review, as usually we found at the end of abstract and introduction. Please include a short sentence at the end of the sections cited to clarify this aspect.

2) The references start in the abstract, while usually the abstract not reported reference. Sincerely I don’t know if in this specific journal permit tha,  but I suggest to remuve references from the abstract.

3) Line 59 As you intend for real time, Please clarify.

4) The wearable devices have one very important role to developed a real peripherical and disseminated telemedicine. I suggest to spent some sentence in more about that, describing the evolution of that.

5) Results, all the first part of the results are NOT results but method (line 141-142) and background (line 143-152) please move this important information in the correct sections.

Author Response

Author’s Reply to the Review Report (Reviewer 1) 

Respectful Reviewer, 

Thank you for allowing us to submit a revised manuscript for further consideration. We appreciate the your time and effort dedicated to providing valuable feedback on our manuscript. We are grateful for the profound reading and detailed comments on our paper. We have been able to incorporate changes to reflect most of the suggestions. We rewrote the paper based on the previous version. Below please find a point-by-point response to the your comments and concerns. 

1) I fatigue to find a clear description of the scope of the review, as usually we found at the end of 

abstract and introduction. Please include a short sentence at the end of the sections cited to clarify this aspect.  

Author’s response: Thank you for the valuable suggestion. We rewrote the abstract so it is more clear. 

2) The references start in the abstract, while usually the abstract not reported reference. Sincerely I 

Don't know if in this specific journal permit tha,  but I suggest to remuve references from the abstract.  

Author’s response: Thank you for the remark. The references were removed from the abstract.  

3) Line 59 As you intend for real time, Please clarify.  

Author’s response: The real-time meaning was synchronous or direct face-to-face interactive consultation between a patient and consultant.  

4) The wearable devices have one very important role to developed a real peripherical and 

disseminated telemedicine. I suggest to spent some sentence in more about that, describing the 

evolution of that.  

Author's response: Thank you for this suggestion. We added a few sentences abou the wereable devices. 

5) Results, all the first part of the results are NOT results but method (line 141-142) and background 

(line 143-152) please move this important information in the correct sections.  

Author's response: We changed the structure of the article, and the introduction, material and methods, results, and conclusions contain relevant issues.  

Reviewer 2 Report

In this paper, the authors briefly discuss the key players in the telemedicine industry, three key technological trends, as well as the advantages (greater patient convenience, lower costs) and disadvantages (legal requirements, financial barriers, business strategies, human resources) of the industry. They cover the current issues in the United States that other nations developing telemedicine policies may encounter in the future.

I put my observations as below:

1.     The abstract does not accurately represent the study. It should be updated.

2.     The abstract and the remainder of the article include no preface. Please supply.

3.     The objective and context of this work are not explained in the introductory section. A paragraph about paper organisation should be included in the Introduction section.

4.     The study's purpose/objectives, rationale, and importance are all absent.

5.     The methods section is incorrect. I couldn't figure it out.

6.     The results section is just a representation of raw data.

7.     In the Discussion section, there is no discussion of any issue. The discussion part seems to be a literature review.

8.     It is advised that future research directions be provided.

9.     The research has no bounds.

Author Response

Author's Reply to the Review Report (Reviewer 2) 

Respectful Reviewer, 

Thank you for allowing us to submit a revised manuscript for further consideration. We appreciate the your time and effort dedicated to providing valuable feedback on our manuscript. We are grateful for the profound reading and detailed comments on our paper. We have been able to incorporate changes to reflect most of the suggestions. We rewrote the paper based on the previous version. Below please find a point-by-point response to the your comments and concerns. 

"I put my observations below:  

  1. The abstract does not accurately represent the study. It should be updated. 

Author's response: Thank you for this remark. We rewrote the abstract that it reflects the whole study.  

  1. The abstract and the remainder of the article include no preface. Please supply. 

Author's response: We are thankful for the valuable suggestion. We made the indicated changes to the main text. 

  1. The objective and context of this work are not explained in the introductory section. A paragraph about paper organisation should be included in the Introduction section. 

Author's response: We made the changes in the Introduction. Additionally, material and methods, results, and discussion sections were rewritten to clarify the manuscript.  

  1. The study's purpose/objectives, rationale, and importance are all absent. 

Author's response: We made more effort to make the rewritten manuscript more logical and consistent. The discussion was focused on some aspects of telemedicine, like organizations in the field of telemedicine, main aspects descriptions from the example out of seven annual conferences on Telemedicine and eHealth in Poland, and finally, some discussion on the main directions and technological benefits and challenges of telemedicine to meet the target of the special issue "Pattern Recognition and Medical Data Analytics in Telemedicine" of the journal.  

  1. The methods section is incorrect. I couldn't figure it out. 

Authors' response: We rewrote the methods section so it is clearer. 

  1. The results section is just a representation of raw data. 

Authors' response: Thank you for this comment. The results currently shall represent the study results with the historical view of local (national) telecommunication development as the environment for the further rise of telemedicine initiatives.  

  1. In the Discussion section, there is no discussion of any issue. The discussion part seems to be a literature review. 

Author's response: Thank you for the suggestion. We agree; this is a summary and invitation to the discussion stage of the article, aimed at referring to the subject with which research should appear, the results of which will be presented in an article for Applied Sciences, special issue "Pattern Recognition and Medical Data Analytics in Telemedicine.  

  1. It is advised that future research directions be provided. 

Author's response: Thank you for the valuable suggestion. We provided some additional remarks in the conclusions.   

  1. The research has no bounds. 

Author's response: We hope that the conference we refer to in the manuscript will also be a mirror for monitoring trends in telemedicine, not only in Polish but also globally in the coming years. 

Reviewer 3 Report

Thank you for invited me. In my opinion, although this article has some obvious points that can be deleted, as a whole is a good resource about the basic and fundamental concepts of telemedicine. It can be useful for whom interested to telemedicine field. I find this article worthy to read and I think can be accepted as this format. Thank you

Author Response

Author's Reply to the Review Report (Reviewer 3) 

Respectful Reviewer, 

Thank you for allowing us to submit a revised manuscript for further consideration. We appreciate the your time and effort dedicated to providing valuable feedback on our manuscript. We are grateful for the profound reading and detailed comments on our paper. We have been able to incorporate changes to reflect most of the suggestions. We rewrote the paper based on the previous version. 

Report Review 3  

Thank you for invited me. In my opinion, although this article has some obvious points that can be deleted, as a whole is a good resource about the basic and fundamental concepts of telemedicine. It can be useful for whom interested to telemedicine field. I find this article worthy to read and I think can be accepted as this format. Thank you  

We would like to express our sincere gratitude for the time you spend reviewing our manuscript. We hope the changes we made will show improvement. We also hope that the revised version of the manuscript is now improved and satisfactory. 

Round 2

Reviewer 2 Report

It has been improved. I'm satisfied.

Thanks